



# Determining stages of cirrus life-cycle evolution: A cloud classification scheme

Benedikt Urbanek[1], Silke Groß[1], Andreas Schäfler[1], and Martin Wirth[1]

[1]Deutsches Zentrum für Luft- und Raumfahrt, Institut für Physik der Atmosphäre, Oberpfaffenhofen, Germany

*Correspondence to:* Benedikt Urbanek (benedikt.urbanek@dlr.de)

**Abstract.** Cirrus clouds impose high uncertainties on climate prediction, as knowledge on important processes is still incomplete. For instance it remains unclear how cloud microphysical and radiative properties change as the cirrus evolves. Recent studies classify cirrus clouds into categories including "in situ", "orographic", "convective" and "liquid origin" clouds and investigate their specific impact. Following this line, we present a novel scheme for the classification of cirrus clouds that addresses the need to determine specific stages of cirrus life-cycle evolution. Our classification scheme is based on airborne Differential Absorption and High Spectral Resolution Lidar measurements of atmospheric water vapor, aerosol depolarization, and backscatter, together with model temperature fields and simplified parameterizations of freezing onset conditions. It identifies regions of supersaturation with respect to ice (ISSR), heterogeneous and homogeneous nucleation, depositional growth, and ice sublimation and sedimentation with high spatial resolution. Thus the whole cirrus life-cycle can be traced. In a case study of a gravity lee wave influenced cirrus cloud, encountered during the ML-CIRRUS flight campaign, the applicability of our classification is demonstrated. Revealing the structure of cirrus clouds, this valuable tool might help to examine the influence of life-cycle stages on the cloud's net radiative effect and to investigate the specific variability of optical and microphysical cloud properties in upcoming research.

## 1 Introduction

Cirrus play an important role for weather and climate: besides their influence on the water vapor budget in the upper troposphere through condensation and evaporation (Dinh et al., 2014) and dynamics due to latent heat (Spichtinger, 2014), they modify the radiation balance of the Earth and atmosphere. Thin, opaque cirrus clouds transmit most of the incident solar radiation and absorb long-wave radiation from the Earth's surface. As they are typically high and cold, they only emit little long-wave radiation into space, and thus cause a trapping of radiative energy in the Earth-atmosphere system, which eventually contribute to a rising surface temperature. If the cloud is thick, reflection of solar radiation back to space can get greater than the long-wave absorption, and consequently can cause the surface of the Earth to cool (Baran, 2009). This net radiative effect depends on macroscopic cloud properties like optical thickness, ice water content, and geometric extent as well as on its microphysical parameters such as ice crystal number, size, and shape (Schnaiter et al., 2016; Gallagher et al., 2012; Zhang et al., 1999).

Today many factors are known that determine these properties: the amount and composition of natural and anthropogenic aerosol particles in the troposphere and their ability to nucleate ice crystals (DeMott et al., 2010), the exact freezing condition





and mechanism (Cziczo et al., 2013), updraft velocity during cloud formation (Kärcher and Lohmann, 2002), water vapor supersaturation (Krämer et al., 2009), cloud age (Korolev and Isaac, 2006), etc. In idealized settings, there is also a good understanding of the relevant processes in cloud formation and break up. But in nature no two clouds are alike and there exists a confusing variability of conditions under which they occur. This makes it difficult to represent cirrus clouds adequately in

5 global circulation models for weather and climate prediction (Baran, 2012; Zhang et al., 2005).

 In order to gain more insight into the particular role of different cirrus clouds, great efforts were made to classify cirrus by the meteorological contexts in which they occur (Jackson et al., 2015; Muhlbauer et al., 2014). Categories include "synoptic", "orographic", "lee wave" and "anvil" cirrus. Recently Krämer et al. (2016) introduced a more general classification distinguishing the groups of "liquid origin" and "in situ" clouds that describe whether the cirrus formed from a pre-existing liquid

10 cloud or from cloud-free air. Such a classification of recorded data is a prerequisite for statistically investigating the specific properties and influences of different clouds, and to extract the governing mechanisms and parameters from remote sensing and in situ measurements.

 Likewise, detailed knowledge of cloud properties at different stages of evolution is yet to be gained, as a cloud is expected to show different properties at the time of formation and break up. During the evolution of a cloud ice particle number, effective

15 radius and particle shape evolve as well. For instance, Korolev and Isaac (2006) showed that during dissipation, ice crystals loose single facets and corners, thus changing their geometry significantly. Besides such changes of microphysical properties, also macrophysical qualities like the exact location, altitude and extent of cloud parts in a specific evolution state may result in a different net radiative effect.

 A statistical study (Comstock et al., 2004), evaluating ground-based Lidar data taken in France over a time period of one year,

20 and in situ measurements (Korolev and Isaac, 2006; Heymsfield and Miloshevich, 1995; Heymsfield, 1975) indicated that there is a vertical order of cirrus life-cycle stages with ice nucleation near cloud top level, deposition of water vapor onto ice crystals and thus particle growth in the middle, and sublimation and sedimentation at cloud base level. Due to its statistical origin, this proposed structure must be seen as an averaged reference order, where individual clouds may show strongly differing distributions (Groß et al., 2014). An evolution classification of individual cloud parts facilitates the detailed investigation of all

25 the above mentioned influences.

 Such a classification needs information on relative humidity with respect to ice ($RH_i$) and temperature, as they are two governing variables in ice particle formation, growth, and disappearance (Pruppacher and Klett, 2010). In order that ice can form in the atmosphere, $RH_i$ must reach or surpass 100 %. It is well known that tropospheric air masses often show substantial supersaturations with respect to ice (Spichtinger et al., 2003). These so called ice-supersaturated regions (ISSR) result mainly

30 from upward motion of air masses originating in diverse atmospheric dynamics like large scale synoptic ascends (e.g. warm conveyor belt), convective systems or meso-scale gravity waves (Spichtinger et al., 2005a, b; Field et al., 2001).

 The existence of an ISSR does not automatically imply the existence or formation of a cirrus cloud. For the homogeneous freezing (HOM) of solution droplets at cirrus temperatures (typically below 235 K), high supersaturations in the order of 140 % are necessary (Koop et al., 2000). Additionally, solid aerosol particles can act as ice nuclei and thus lead to freezing under a





much broader range of conditions. Heterogeneous freezing onset temperatures and saturation ratios depend strongly on aerosol type, coating of the particles, and their size and are still subject to current research (Hoose and Möhler, 2012).

Once ice particles are present, remaining supersaturation is depleted by deposition of water vapor onto existing crystals. Depending on the particle number and average radius, it may take a few minutes to a few hours for the equilibrium of $100\,\%$

to be reached (Korolev and Mazin, 2003). Furthermore, effects that hinder phase relaxation are discussed, stemming from e.g. liquid coating around ice particles (Bogdan and Molina, 2009), or dynamic processes (Spichtinger and Krämer, 2013). Thus, supersaturation inside of ice clouds can persist for a substantial amount of time. Likewise, regions of subsaturation with respect to ice can emerge when heavy ice particles sediment out of the ISSR, or $RH_i$ is reduced by warming. These regions are dominated by sublimation of ice crystals (Korolev and Isaac, 2006; Kübbeler et al., 2011).

It can be seen that detailed knowledge of humidity and temperature in and around the cloud, as well as knowledge about freezing onset conditions is necessary in order to identify different stages of cloud evolution. The airborne Differential Absorption Lidar WALES ("WAter vapour Lidar Experiment in Space"), flown aboard the German research aircraft HALO ("High Altitude and LOng range", model: Gulfstream G550), makes major parts of the needed data available. It provides an unique dataset of collocated, high spatial resolution measurements of atmospheric backscatter, depolarization and water vapor, en-

abling us to distinguish in-cloud and cloud-free regions, to identify the relevant aerosol type in the vicinity of the cloud and to calculate relative humidity.

In this paper, we present a detailed classification scheme for the evolution of cirrus clouds. Besides WALES measurements we use complementary model temperature fields from the European Centre for Medium-range Weather Forecasts (ECMWF). Provided with high resolution, two-dimensional Lidar cross-sections of the atmosphere, we are able to study the structure

of clouds and the spatial distribution of classified evolution stages. We apply our scheme in a first case study of a lee wave influenced cirrus cloud over France, encountered during the ML-CIRRUS 2014 campaign (Voigt et al., 2016), demonstrate its applicability, and investigate the impact of both small-scale and large-scale dynamics on the cloud structure. We end with a summary of the classification scheme and a brief outline of its potential in cirrus cloud research.

## 2  Water vapor remote sensing during ML-CIRRUS

In spring 2014, the Mid-Latitude Cirrus experiment ML-CIRRUS was conducted. It was designed to investigate natural cirrus and anthropogenic contrail cirrus with regard to their nucleation, life-cycle, and climate impact. In this campaign, the German research aircraft HALO, equipped with a combined in situ and remote sensing payload, performed 16 measurement flights above Europe. The on-board cloud probes, WALES Lidar and novel ice residual, aerosol, trace gas and radiation instruments probed mid-latitude cirrus clouds originating from e.g. air traffic, warm conveyor belts, jet streams, or mountain waves (Voigt

et al., 2016).

WALES is an airborne Differential Absorption Lidar that measures the tropospheric water vapor concentration below the research aircraft by simultaneously emitting laser pulses at three online and one offline wavelength in the water vapor absorption band around 935 nm (Wirth et al., 2009). The averaged pulse energy is 35 mJ with a repetition rate of 200 Hz. The three





online wavelengths provide the needed sensitivity to compose a complete water vapor profile from the partly overlapping line contributions that ranges from just below the aircraft down to ground level. Additionally, WALES is equipped with one channel at 1064 nm and one High Spectral Resolution channel at 532 nm using an iodine filter. Both receiver channels are designed to detect the depolarization of the backscattered light (Esselborn et al., 2008).

WALES is capable to provide collocated measurements of humidity in the form of water vapor volume mixing ratio $r_w$, backscatter ratio (BSR), and aerosol depolarization ratio (ADEP). Those measurements form a two dimensional curtain along the flight track of the research aircraft intersecting the atmosphere below. The Lidar data we use in this paper has a vertical resolution of 15 m. Raw data is sampled at a rate of 5 Hz. At HALO's typical ground speed of 210 ms$^{-1}$ and after averaging for a better signal-to-noise ratio, horizontal resolution is 2.5 km for humidity and 210 m for BSR and ADEP.

We use ECMWF analysis and forecast temperature data that we interpolate both temporally and spatially onto the Lidar measurement cross-section. Then we calculate relative humidity with respect to ice from this temperature information and the measured absolute humidity:

$$RH_i = \frac{r_w \cdot n_{air} \cdot T \cdot k_B}{e_{sat,i}(T)}, \tag{1}$$

with temperature $T$, volume number density of air $n_{air}$, and Boltzmann constant $k_B$. We use the parameterization for water

vapor saturation pressure over ice $e_{sat,i}$ by Murphy and Koop (2005).

The accuracy of calculated relative humidity relies strongly on the quality of absolute humidity and temperature data. WALES humidity measurements exhibit a mean statistical uncertainty of 5 %. The applicability of ECMWF temperature in this calculation was investigated by Groß et al. (2014). They showed that during ascent and descent of a similar research flight in 2010 the temperature difference between ECMWF and on-board temperature sensors was always less than 1 K and

estimated a resulting maximum relative uncertainty of 10-15 % in the calculated $RH_i$ at typical cirrus temperatures.

## 3    Cirrus evolution classification scheme

With atmospheric lidar cross-sections at hand, we are able to identify in-cloud and cloud-free regions by applying a threshold for the backscatter ratio (see Fig. 1). As there is no sharp boundary between a cloud and its surrounding, this threshold value holds a certain arbitrarity. In the case study, we use a value of 2, but in cases where e. g. thick aerosol layers are present this

threshold might need to be increased to avoid classifying parts of the aerosol layer as in-cloud regions.

Looking at cloud-free parts of the cross-section, regions that might possibly lead to cirrus cloud formation can be identified by searching for data points exhibiting ice supersaturation ($RH_i > 100$ %). Moderately supersaturated cloud-free parts are classified as ISSR. With higher supersaturations, the chances for the imminent nucleation of ice particles get increasingly higher. Therefore we introduce the classes HET$_{out}$ and HOM$_{out}$ in our classification. They represent regions outside of the

cloud where onset condition for heterogeneous and homogeneous freezing get surpassed, respectively. Their classification is implemented via temperature dependent humidity thresholds.

It should be noted that ice is forming as soon as conditions for homogeneous freezing get reached, as there is always a sufficient amount of solution droplets in the atmosphere. Therefore, a cloud classification should not feature considerable



regions of $\text{HOM}_\text{out}$. This fact should be kept in mind when choosing a BSR threshold value for the cloud border detection, making sure that HOM regions lie inside the cloud. $\text{HET}_\text{out}$ regions, however, may exist in cases with no sufficient amount of aerosol ice nuclei. For homogeneous freezing, we extract a parameterization of the temperature dependent onset humidity for HOM from Koop et al. (2000, their Fig. 3) for a nucleation rate $\omega = 0.0167\ \text{s}^{-1}$ and a droplet size of 0.5 μm (see Table 1,

$RH_{i,HOM}(T)$).

To determine a humidity threshold for HET, detailed information of the involved aerosol type, its coating, and size distribution would be required. Then results from laboratory experiments on onset freezing temperatures and saturations for this kind of aerosol could be used. As heterogeneous freezing conditions are still subject to current research (Hoose and Möhler, 2012) and as comprehensive aerosol information is difficult to acquire solely from remote sensing, we make only a coarse distinction

between two important aerosol types: mineral dust (MD) and coated soot (CS). Together with a synergistic analysis, WALES measurements of aerosol linear depolarization ratio (ADEP) and Lidar ratio can be used to identify the relevant aerosol type in the measurement area. To this end, we use an aerosol classification suggested by Groß et al. (2013). Then we employ simplified onset parameterizations $RH_{i,HET}^{MD}(T)$ and $RH_{i,HET}^{CS}(T)$ (see Table 1 and Krämer et al. (2016, their Fig. 4)). Until more detailed parameterizations are available, this imposes a uncertainty for the determination of the exact border of heterogeneous

freezing regions. The classes ISSR and $\text{HET}_\text{out}$ represent pre-stages of Cirrus formation and indicate regions where a cirrus cloud is likely to develop.

Inside of a cloud (BSR > 2), we proceed in the same manner. When the $RH_{i,HOM}(T)$ threshold is surpassed we classify as $\text{HOM}_\text{in}$. This region shows active ice nucleation. Together with $\text{HET}_\text{in}$, that we classify analogously, it represent the youngest evolution stage of a cirrus cloud. $\text{HET}_\text{in}$ will also show active nucleation as long as ice nuclei are present in the freezing region.

When relative humidity inside the cloud is lower than the freezing thresholds, we classify as DEP, as the remaining supersaturation is depleted by deposition of water vapor onto the existing ice particles. This intermediate evolution stage is dominated by depositional growth of ice crystals. The final evolution stage of a cloud sets in, when relative humidity falls below 100 %. In such an environment ice inevitably must sublimate. We classify this region as SUB.

This classification scheme is applied independently to every recorded data point, enabling the detailed study of individual

cloud parts.

## 4   Case study ML-CIRRUS 2014-03-29

We demonstrate the applicability of our classification scheme in a cirrus case that was obtained during the ML-CIRRUS field campaign on 29 March 2014. The meteorological situation over Western Europe and the Iberian Peninsula on the flight day is dominated by a trough extending from west of Ireland to the Iberian Peninsula and further to the western part of North Africa

(Fig. 2 a). At 300 hPa, high southerly winds with wind speeds up to 35 ms$^{-1}$ are observed on the leading edge over Southern France and Spain. Model forecasts indicated the existence of cirrus forming from high updrafts over the Pyrenees, as well as cirrus influenced by lee waves north of the mountain ridge. Additionally high dust concentrations of Saharan mineral dust were expected.



In this meteorological setting the research flight was performed with the aim to sample all stages of cirrus evolution that resulted from an overflow of the Pyrenees with high wind speeds and consequent gravity wave excitations in the lee of the mountain ridge. Therefore the flight path in the relevant measurement region was chosen to run along the main wind direction, sampling the clouds along their path of advection.

The flight (Fig. 2, red flight path) started in Oberpfaffenhofen, Germany at 12:37 UTC and first went westward towards Paris, followed by a southward flight leg towards Spain at an altitude of 11200 m. The investigated cirrus cloud was encountered over Southern France during this leg that is running with a bearing of 190° (white flight leg). Inside cirrus clouds, over the Pyrenees mountains, three legs at different lower altitudes followed. From the Mediterranean coast the aircraft turned eastward and probed cirrus at several altitudes near the Balearic islands before it went northward towards Oberpfaffenhofen (landing at 19:50 UTC).

## 4.1 Cirrus leg overview

The following discussion of the classification scheme focuses on the southward flight leg stretching about 400 km to the north and 200 km to the south of the Pyrenees (Fig. 2, white flight path). Fig. 2 b shows a false color image of the Pyrenees area derived from SEVIRI ("Spinning Enhanced Visible and InfraRed Imager") data at 14:30 UTC. Higher, cooler clouds have a bluish color; low clouds are depicted in yellow. Coming from the North, the flight path intersects an ice cloud that is part of a larger cloud regime expanding from Southern France towards the Algerian coast. This cloud is followed by a completely cloud-free area north of the Pyrenees. Over the mountain ridge some localized high clouds are crossed.

In Fig. 3 we plot a cross-section showing backscatter ratio at a wavelength of 532 nm along the chosen part of the flight path. Here atmospheric features apparent in Fig. 2 can be studied in greater detail. On the lee side, north of the Pyrenees (14:19-14:34 UTC), a high cirrus cloud is observed that extends from a height of 7 km to 11 km. The southern and middle parts are dominated by high BSR values from 50 up to 200, whereas the northern section exhibits lower values. Aerosol linear depolarization ratios of more than 30 % inside the cloud (not shown) and temperatures below 240 K clearly indicate a pure ice cloud. Over the Pyrenees (14:42-14:53 UTC) a lower cirrus cloud is located at an altitude of about 6 km. Its spatially restricted occurrence over the mountain ridge indicates a formation due to forced updrafts stemming from the southerly cross-mountain flow. Even lower, at a height of 4 km a thick aerosol layer is discernible. An analysis of ADEP indicates that this layer contains Saharan mineral dust which is consistent with the origin of the air masses in North Africa.

Furthermore, in the region between the two clouds (14:34-14:43 UTC) gravity lee waves are discernible at an altitude of about 9500 m and also in the lower aerosol layer. These waves are expected to influence at least parts of the northern cirrus cloud. We will investigate them in more detail in Sect. 4.3.

## 4.2 Classifying evolution stages

In the following we will apply our classification scheme to the high cirrus cloud north of the Pyrenees. Fig. 4 and Fig. 5 give a close-up view of the selected data marked with a black rectangle in Fig. 3. Water vapor volume mixing ratio $r_w$ measured by WALES is plotted in Fig. 4 a. A black contour line (BSR= 2) marks the cloud border. Being an absolute humidity measure,





$r_w$ generally decreases with increasing altitude, as temperature is decreasing and almost all sources of water vapor are located at the Earth's surface. Contrastingly, a humid layer, surrounded by dryer air at a height of approximately 9000 m, can be found upstream of the cirrus cloud (14:34 – 14:40 UTC). In this region, the water vapor data exhibits the same oscillations as previously seen in the BSR data.

Relative humidity with respect to ice (Fig. 4 b) is calculated from this data using the ECMWF model temperature field. As expected, supersaturated regions (blue) are found mostly inside of the cirrus. There are also major subsaturated regions (red) in the northern part of the cloud. South of the cirrus high supersaturations exist in cloud-free air, mostly in the crests of the gravity waves (14:34-14:36 UTC). The highest supersaturations are measured in the most southern part of the cloud (14:33-14:34 UTC). They indicate a nucleation region.

To investigate individual parts of the cloud in more detail, we apply our classification and visualize the result in Fig. 5. Data pixels are classified (Sect. 3.2) and marked in color accordingly and in-cloud and cloud-free regions can be distinguished by the black contour line for a BSR value of 2. Heterogeneous freezing is identified using the $RH_{i,HET}^{MD}(T)$ threshold. Subsaturated regions outside of the cloud are left blank and areas where no valid data is available are indicated by black hatching.

The above mentioned humid layer, discernible in Fig. 4, reaches ice supersaturation (ISSR) in the two crests of the gravity
lee wave to the south of the cloud (14:34-14:36 UTC). Here, values of $RH_i$ are even higher than the threshold for HET freezing. At the cloud edge, also the HOM freezing threshold is surpassed (14:33-14:34 UTC). The southern section of the cirrus (14:32-14:34 UTC) is dominated by ice nucleation and represents the youngest part of the cloud.

In the middle (14:26-14:32 UTC), a section of moderate supersaturation (DEP) is located. This is an already well developed part of the cirrus that is dominated by depositional growth of ice crystals. After an initial ascent (14:32-14:34 UTC), the cloud
top level slopes from over 10 km down to under 9 km at the northern edge. This indicates a large-scale descent reducing supersaturation and evoking the intermediate DEP region as well as large connected regions of subsaturation (SUB) in the northern part of the cloud (14:19-14:26 UTC). Here the cloud is starting to break up, as ice particles are sublimating.

From these results all cirrus life-cycle stages can be identified: from ice nucleation (HET, HOM) aided by vertical displacements in a gravity lee wave, to crystal growth by deposition of water vapor in a moderately supersaturated region DEP, to the
dissolving of the cloud in a subsaturated region (SUB), probably caused by a large-scale descent.

The detailed distribution of these major stages of cirrus evolution features a horizontal order instead of a general vertical structure found in cirrus clouds over France (Comstock et al., 2004). Surprisingly we even find SUB regions at the cloud top level located above DEP regions in the northern part of the cloud (Fig. 5). Similarly, model simulations, investigating the influence of dynamics on the evolution of a cirrus cloud (Spichtinger and Gierens, 2009), also found more complex horizontal
distributions deviating from a simplistic cirrus evolution pattern comprising ice nucleation at cloud top level, a crystal growth in the middle an sublimation at the bottom. Thus our classification illustrates how the large-scale meteorological context, wind and gravity wave fields can effect the structure of individual clouds.



### 4.3 Investigating the influence of lee waves

A special feature of this case study is the presence of lee wave patterns in the cloud region. Fig. 6 gives a close-up view of BSR data in the cloud-free area south to the cirrus cloud. A layer of slightly higher BSR (> 1.2) is located above an altitude of 9500 m. It shows clear oscillations at its boundary to a lower, cleaner layer of air. One period extends over about 66 seconds in

measurement time which corresponds to an apparent wavelength of 14 km with vertical displacements of up to 190 m.

ECMWF model data, available for this cirrus case, features a horizontal grid spacing of about 16 km and thus is not able to resolve the small-scale lee waves in its temperature gradients. As a result, temperatures in the crests might be even lower and $RH_i$ values therefore underestimated.

To investigate this possible deviation and its influence on our classification results, we simulate adiabatic cooling of an air

parcel along a hypothesized trajectory (Fig. 5, blue line) in front of the cirrus cloud. The trajectory runs 200 m under and parallel to the contour line of BSR = 1.2 (not shown), that separates the two distinct layers of air. The wind direction in this region differs by less than 10° from the flight path. This makes us confident that the simulated trajectory resembles a real trajectory reasonably well, under the assumption of a stationary air flow.

SEVIRI images from 14:00 UTC and 15:00 UTC (not shown) indicate, that the northern and southern edges of the cloud are

15 moving only about 20 km to the north along the flight path. That corresponds to a cloud velocity of under 6 ms$^{-1}$, compared to wind speeds of up to 35 ms$^{-1}$ at cirrus altitude. As in a lee wave cloud (Field et al., 2012), air is flowing through the region, becoming part of the cloud in the south and leaving the cloud in the north. In the confined area and time frame of our simulation, we consider the underlying wind and wave fields to be quasi-stationary. However this might certainly not be true for the duration of the whole flight leg (14:18 - 14:41 UTC). Also our simulation is not intended to provide corrected temperature

data but to illustrate the general influence of gravity waves on cloud formation and of unresolved temperature fluctuations on our classification.

In Fig. 7, ECMWF temperature and relative humidity calculated with ECMWF temperature along the trajectory are plotted (blue) as a function of measurement time. As the trajectory follows the vertical displacements of the gravity wave along the wind direction, i. e. from right to left in Fig. 3 to 7, relative humidity and ECMWF temperature show oscillations. They stem

from the undisturbed temperature gradient, as ECMWF does not resolve the small-scale lee waves. Besides clear oscillations, a development towards higher $RH_i$ and lower temperature, approaching the cloud, is discernible. From 14:36:20 UTC on, $RH_i$ shows supersaturation and reaches values of 120 % and 130 % in the following two crests, surpassing the HET threshold.

Now we start a parcel at the beginning of the trajectory (measurement time: 14:40 UTC) initialized with the ECMWF temperature at this point. As it follows the trajectory, the temperature is calculated from its vertical displacement using the dry

adiabatic temperature gradient. The simulated temperature and relative humidity is plotted in green. Compared to ECMWF, the temperature in the last two crests south of the cloud edge is more than 0.5 K lower and values of $RH_i$ are higher by 10 %. The deviations result from the non-adiabatic temperature gradient in the ECMWF data.

These results emphasize the role of lee waves in the most southern part of the studied cirrus cloud. Comparably cooler temperatures due to adiabatic cooling in the wave crests facilitate the early nucleation of ice crystals. We find that our classifi-



cation, using original ECMWF data with relatively coarse spatial resolution (horizontal grid spacing: 16 km), is able to reveal the relevant classification features within the gravity wave region. The classification quality will be even better in cases where latest ECMWF data with an improved grid spacing of 9 km, or output from regional models is available.

Overall our classification proved to be applicable in a meteorological context that comprises both small-scale and large-scale dynamics. It identifies all relevant stages of cirrus evolution and their detailed spatial distribution and thus, also reveals the influences of gravity waves and large-scale atmospheric motion on the studied cirrus cloud.

## 5    Summary and conclusions

We presented a novel cirrus classification scheme capable of identifying all evolution stages of the cirrus life-cycle. It is based on airborne Lidar measurements with high spatial resolution of water vapor, backscatter and aerosol depolarization. This data is used together with ECMWF model temperature fields and knowledge and assumptions about onset conditions for homogeneous and heterogeneous freezing to retrieve a cross-section of the cloud, revealing the detailed distribution of evolution stages

In cloud-free air (BSR < 2) ice supersaturated regions (ISSR) as well as regions of homogeneous ($HOM_{out}$) and heterogeneous freezing ($HET_{out}$) are determined. These indicate favorable areas for cirrus cloud formation. Inside of a cloud, ice nucleation ($HET_{in}$, $HOM_{in}$), depositional growth (DEP) and sublimation regions (SUB) are distinguished. They represent the formation, growing and break up phase of a cirrus cloud, respectively.

We demonstrated the applicability of our classification in a first case study of a cirrus cloud that was observed in a complex meteorological situation comprising a thick aerosol layer, large-scale dynamics and small-scale gravity lee waves. Here it revealed a non-standard horizontal order of the aforementioned life-cycle stages and helped to identify the influence of underlying wind and gravity wave conditions as well as large-scale dynamics on individual parts of the cloud.

With this valuable tool at hand, we are investigating in our ongoing research the large airborne Lidar data set obtained during the ML-CIRRUS campaign. This classification scheme facilitates the study of the spatial distribution of evolution stages and can be used to set in-situ and other remote sensing data, obtained during the campaign, in perspective to cirrus evolution. By combining those data sources, the specific optical and microphysical properties of different cirrus stages now can be explored. Thus we aim to achieve more detailed insights in radiative properties of cirrus clouds under various formation and life-cycle conditions.

*Acknowledgements.* ML-CIRRUS campaign was mainly funded by Deutsches Zentrum für Luft- und Raumfahrt (DLR) and Deutsche Forschungsgemeinschaft (DFG). This work has been funded by DLR VO-R young investigator group. The authors like to thank the staff members of the HALO aircraft from DLR Flight Experiments for preparing and performing the measurement flights, Prof. Dr. Christiane Voigt, Dr. Andreas Minikin and everybody contributing to the successful planning and execution of ML-CIRRUS, and the European Centre for Medium-range Weather Forecasts (ECMWF) for providing model data. Our special thanks goes to Dr. Florian Ewald for providing SE-VIRI satellite image data and Dr. Klaus Gierens and Benedikt Ehard for fruitful discussions and helpful suggestions that contributed to the quality of this work.



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





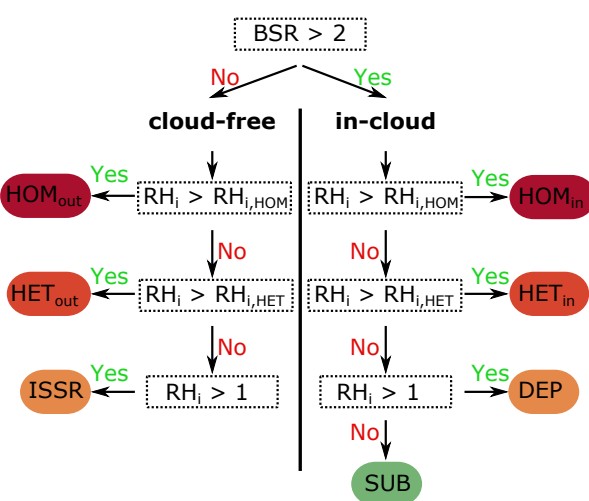

**Figure 1.** Cirrus life-cycle classification scheme based on WALES backscatter ratio (BSR) and relative humidity ($RH_i$) derived from WALES humidity and ECMWF temperature field (description see text).



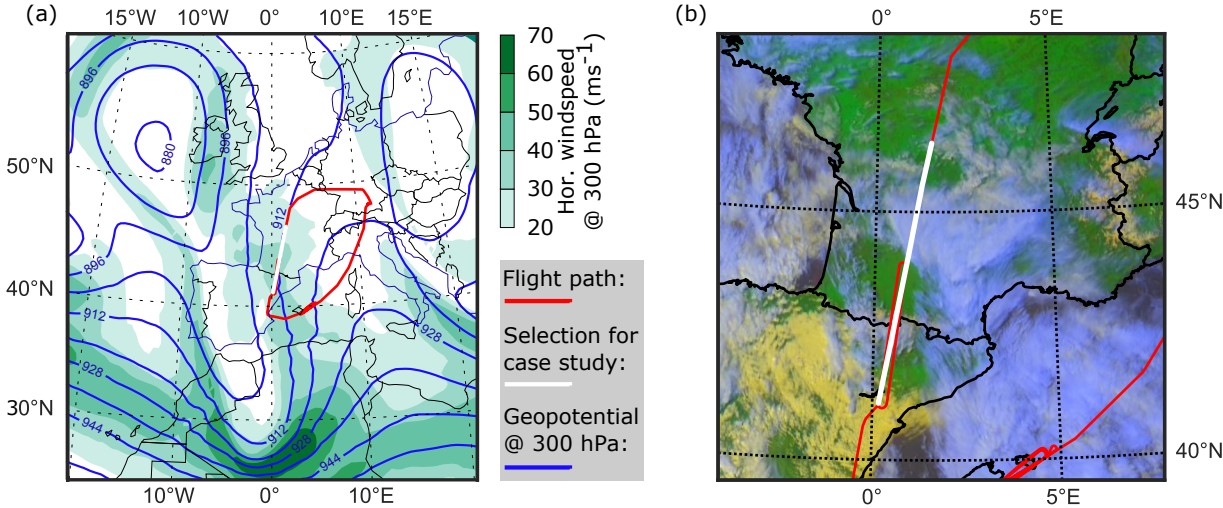

**Figure 2.** Meteorological situation on 29 March 2014 over Western Europe: (a) ECMWF analysis data (12:00 UTC) at 300 hPa of geopotential (blue isolines) and horizontal wind speed (shaded green). (b) SEVIRI false color image taken at 14:30 UTC. High ice clouds have a blue color, lower liquid clouds are yellow. The path of the research flight is plotted in red and the flight leg used in the case study is marked white.





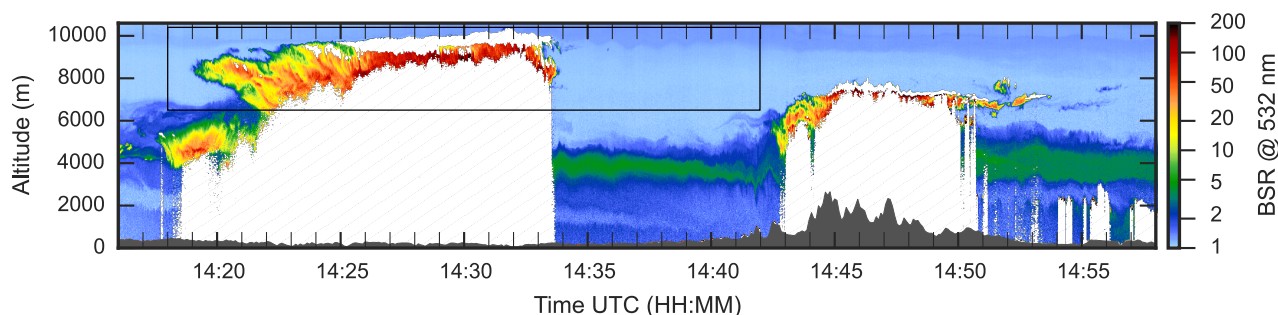

**Figure 3.** Backscatter ratio (BSR) at 532 nm measured along the flight path (white line in Fig. 2). Hatched areas indicate data that was excluded due to detector saturation or low signal to noise ratio and the terrain profile is shown in dark gray. The black rectangle marks the cirrus region that is studied further (see Fig. 4, 5).





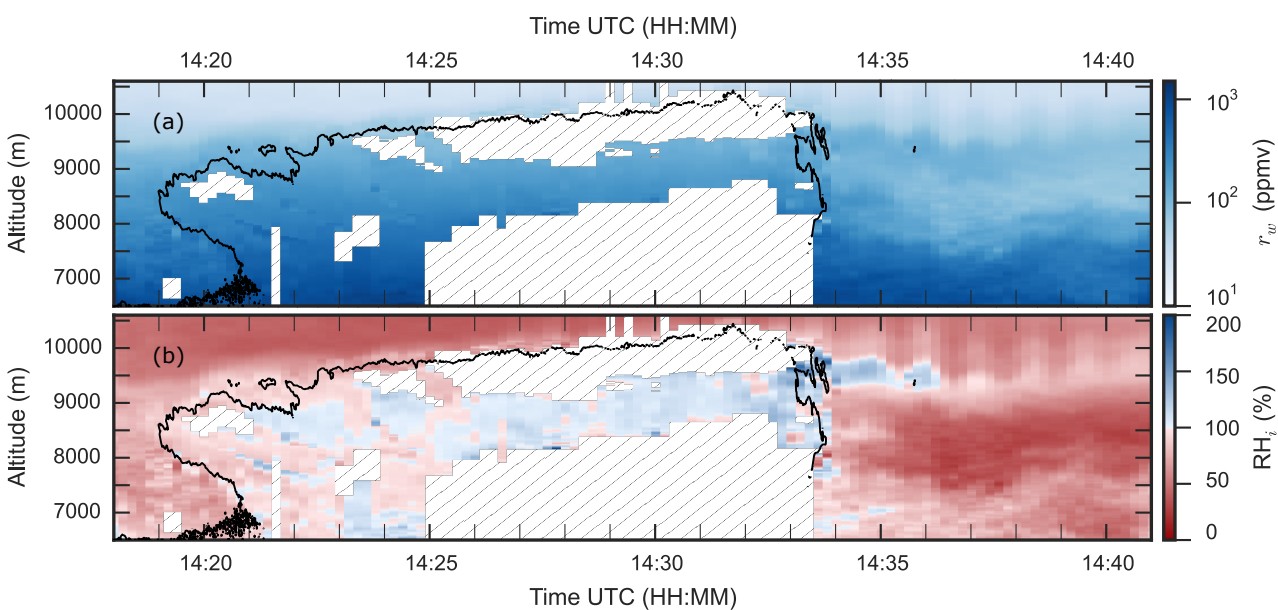

**Figure 4.** Humidity data of cirrus marked in Fig. 3: (a) Water vapor mixing ratio $r_w$ as measured with WALES. (b) Relative humidity with respect to ice $RH_i$ derived from WALES data and ECMWF temperature field. The cirrus is outlined by a black contour line (BSR=2) and invalid data is marked by black hatching.





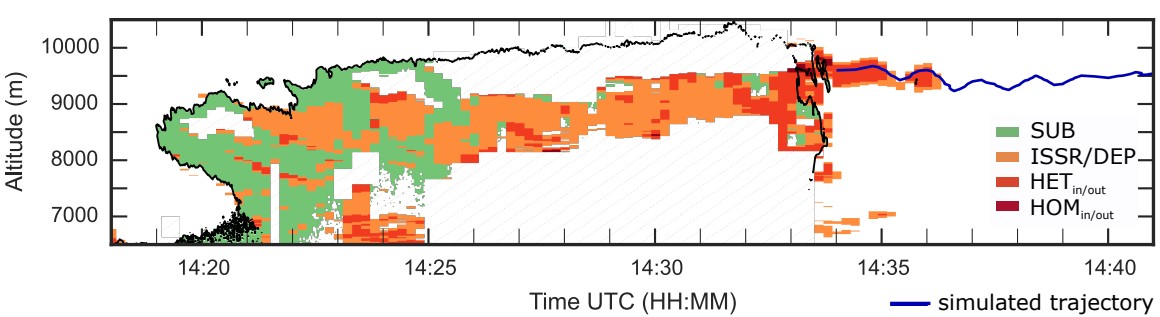

**Figure 5.** Classified life-cycle regions of cirrus cloud. In-cloud and cloud-free regions can be distinguished by the black contour line of BSR = 2. Regions of ice nucleation (HOM$_{in/out}$, HET$_{in/out}$), ice supersaturation outside (ISSR), and depositional growth (DEP) and subsaturation (SUB) inside the cloud are visualized. Subsaturated regions outside the cloud are left blank and invalid data is marked by black hatching. The blue line shows a trajectory used for simulation of adiabatic cooling (see Sect- 4.3 and Fig. 7).



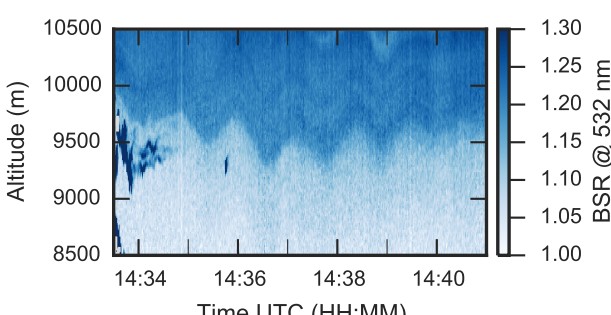

**Figure 6.** Close-up view of BSR data in the lee wave region south to the cirrus cloud.





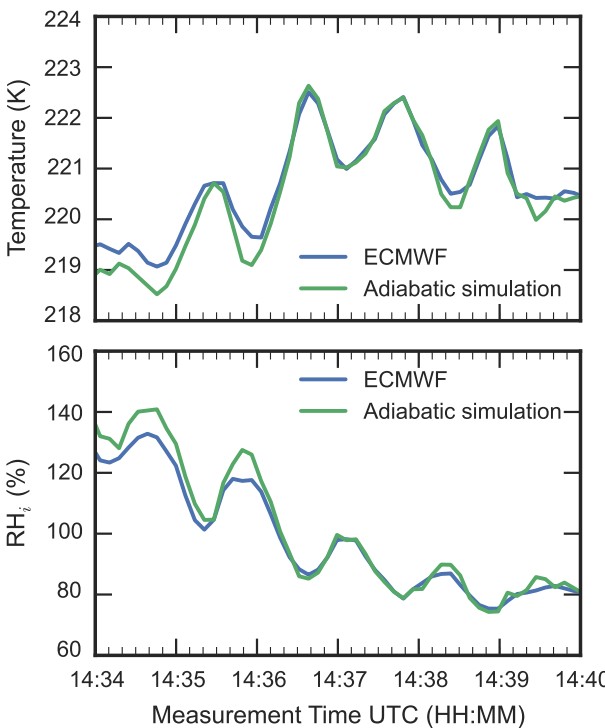

**Figure 7.** Simulation of temperature T and relative humidity $RH_i$. Green lines are simulated values along a derived trajectory (Fig. 5) assuming adiabatic cooling and heating within the gravity wave, respectively. In blue are ECMWF values interpolated to the trajectory location.

**Table 1.** Freezing onset humidity parameterizations

$$
\begin{aligned}
RH_{i,HOM}(T) &= 237\,\mathrm{K} - 0.4 \cdot T && \text{(Koop et al., 2000)} \\
RH^{MD}_{i,HET}(T) &= 134\,\mathrm{K} - 0.1 \cdot T && \text{(Krämer et al., 2016)} \\
RH^{CS}_{i,HET}(T) &= 230\,\mathrm{K} - 0.4\bar{3} \cdot T && \text{(Krämer et al., 2016)}
\end{aligned}
$$