# Peer review of "Determining stages of cirrus life-cycle evolution: A cloud classification scheme"

_Atmospheric Measurement Techniques, 2016_

## Referee Comment (RC1) · Anonymous Referee #1 · 22 Nov 2016

The comment was uploaded in the form of a supplement:
http://www.atmos-meas-tech-discuss.net/amt-2016-332/amt-2016-332-RC1-supplement.pdf

Review of

**Determining stages of cirrus life-cycle evolution:**
**A cloud classification scheme**

by Urbanek et al.

The authors present an attempt to determine the stages of cirrus life-cycle evolution based on in-cloud RHi measurements performed by the airborne Lidar WALES. Though I like the idea and also find the paper well organized and fluently written, I have a major concern with respect to the proposed cirrus life-cycle classification scheme which I explain in the following. To my opinion this point should be cleared before publishing the manuscript in ACP.

**Major comment**: In the introduction, the authors state:
'*In order to gain more insight into the particular role of different cirrus clouds, great efforts were made to classify cirrus by the meteorological contexts in which they occur (Jackson et al., 2015; Muhlbauer et al., 2014). Categories include "synoptic", "orographic", "lee wave" and "anvil" cirrus. Recently Krämer et al. (2016) introduced a more general classification distinguishing the groups of "liquid origin" and "in situ" clouds that describe whether the cirrus formed from a pre-existing liquid cloud or from cloud-free air. Such a classification of recorded data is a prerequisite for statistically investigating the specific properties and influences of different clouds, and to extract the governing mechanisms and parameters from remote sensing and in situ measurements.*'

However, the cirrus life-cycle classification scheme presented in the paper holds only for 'in situ' formed cirrus clouds. In the so-called 'liquid origin' cirrus, the meaning of 'SUB' will be similar, but what about the interpreation of 'DEP', HETin and HOMin in case of pre-existing ice? It is very likely that in case of further lifting of a liquid origin cirrus cloud the supersaturation rises to values of DEP, HETin or HOMin (then, a new, homogeneous nucleation event can occur on top of the liquid origin cirrus), but they are at different stages of cirrus evolution than the in situ cirrus.

In a recent publication of Wernli et al. (2016), GRL, the frequencies of occurence of in situ and liquid origin cirrus are analyzed from 12 years of ERA-Interim ice clouds in the North Atlantic region. Wernli et al. found that: 'Between 400 and 500 hPa more than 50% are liquid-origin cirrus, whereas this frequency decreases strongly with altitude (<10% at 200 hPa).'

Thus, it seems to be important that first of all these two types of cirrus can be identified by a cirrus classification scheme before going in the detail of stages of cirrus life-cycle evolution. So I would highly encourage the authors to continue their work by including an analysis of the cirrus origin prior to the investigation of the stages of evolution.

It might be an idea to first perform a trajectory analysis as done by Wernli et al. (2016) and also Luebke et al. (2016) using ECMWF wind fields and determine wether the back trajectory of an observed air parcel stemmed from temperatures warmer than -38C and carried ice when entering the cirrus temperature range. Then, the classification scheme can be applied to both types separatly.

**Fig. 1.**

**Supplement:**

Review of

**Determining stages of cirrus life-cycle evolution:
A cloud classification scheme**

by  Urbanek et al.

The authors present an attempt to determine the stages of cirrus life-cycle evolution based on in-cloud RHi measurements performed by the airborne Lidar WALES. Though I like the idea and also find the paper well organized and fluently written, I have a major concern with respect to the proposed cirrus life-cycle classification scheme which I explain in the following. To my opinion this point should be cleared  before publishing the  manuscript in ACP.

**Major comment**:  In the introduction, the authors state:
'*In order to gain more insight into the particular role of different cirrus clouds, great efforts were made to classify cirrus by the meteorological contexts in which they occur (Jackson et al., 2015; Muhlbauer et al., 2014). Categories include "synoptic", "orographic", "lee wave" and "anvil" cirrus. Recently Krämer et al. (2016) introduced a more general classification distinguishing the groups of "liquid origin" and "in situ" clouds that describe whether the cirrus formed from a pre-existing liquid cloud or from cloud-free air. Such a classification of recorded data is a prerequisite for statistically investigating the specific properties and influences of different clouds, and to extract the governing mechanisms and parameters from remote sensing and in situ measurements.*'

However, the cirrus life-cycle classification scheme presented in the paper holds only for 'in situ' formed cirrus clouds. In the so-called 'liquid origin' cirrus, the meaning of 'SUB' will be similar, but what about the interpreation of 'DEP', HETin and HOMin in case of pre-existing ice? It is very likely that in case of further lifting of a liquid origin cirrus cloud the supersaturation rises to values of  DEP, HETin or HOMin (then, a new, homogeneous nucleation event can occur on top of the liquid origin cirrus), but they are at different stages  of cirrus evolution than the in situ cirrus.

In a recent publication of Wernli et al. (2016), GRL,  the frequencies of occurence of in situ and liquid origin cirrus are analyzed from 12 years of ERA-Interim ice clouds in the North Atlantic region.  Wernli et al.  found that: 'Between 400 and 500 hPa more than 50% are liquid-origin cirrus, whereas this frequency decreases strongly with altitude (<10% at 200 hPa).'

Thus, it seems to be important that first of all these two types of cirrus can be identified by a cirrus classification scheme before going in the detail of stages of cirrus life-cycle evolution. So I would highly encourage the authors to continue their work by including an analysis of the cirrus origin prior to the investigation of the stages of evolution.

It might be an idea to first perform a trajectory analysis as done by Wernli et al. (2016) and also Luebke et al. (2016) using ECMWF wind fields and determine wether the back trajectory of an observed air parcel stemmed from temperatures warmer than -38C  and carried ice when entering the cirrus temperature range. Then, the classification scheme can be applied to both types separatly.

I am aware about that this will be a lot of additional work, but am also convinced that it will be worth the effort to make the study scientifically sound and useful for future investigations.

**Minor comments:**

1) Page 1, line 24 – page 2, line 1:

'*Today many factors are known that determine these properties: the amount and composition of natural and anthropogenic aerosol particles in the troposphere and their ability to nucleate ice crystals (DeMott et al., 2010), ..*'

This statement is much too promising – amount and properties of IN (Ice Nuclei) are not well known until today, in particular in the temeprature range of cirrus clouds. Please correct.

'*... the exact freezing condition and mechanism (Cziczo et al., 2013), updraft velocity during cloud formation (Kärcher and Lohmann, 2002), ...*'

Same here: it is not clear if the work of Cziczo et al. (2013) is globally valid; updraft velocities during cloud formation are theoretically known, but measurements are difficult and rare.

2) Page 2, line 13-14:

'*...a cloud is expected to show different properties at the time of formation and break up.*'

Better 'dissipation'     instead of    'break up'

**3) Page 3, lines 3-5:**

'*Once ice particles are present, remaining supersaturation is depleted by deposition of water vapor onto existing crystals. Depending on the particle number and average radius, it may take a few minutes to a few hours for the equilibrium of 100 % to be reached (Korolev and Mazin, 2003).*'

Korolev and Mazin (2003) show relaxation times to reach the 'dynamical equilibrium' (steady state), which is -in dependence on the updraft- higher than 100%.  Saturation in cirrus is quickly reached as soon as the cooling stops, i.e. when the updraft is zero.  Please correct.

4) Page 4, line 32:

'*It should be noted that ice is forming as soon as conditions for homogeneous freezing get reached, ...*'

Please correct:
... ice is forming **latest** as soon as conditions for homogeneous freezing get reached,...
since heterogeneous freezing starts earlier at lower RHi $\rightarrow$ higher tempeartures.

5) Page 4, line 32 – page 5, line 3:

' *Therefore, a cloud classification should not feature considerable regions of HOMout. This fact should be kept in mind when choosing a BSR threshold value for the cloud border detection, making sure that HOM regions lie inside the cloud. HETout regions, however, may exist in cases with no sufficient amount of aerosol ice nuclei.* '

Have you chosen the threshold BSR? So  that no HOMout occured ?

Also,  HETout can occur in case  RHi is higher than the chosen threshold RHi_HET, not only due to a lack of IN.

6) Page 5, line12:

'*To this end, we use an aerosol classification suggested by Groß et al. (2013).  Then we employ simplified onset parameterizations $RH^{(MD)}_{i,HET}(T)$ and $RH^{(CS)}_{i,HET}(T)$ (see Table 1 and Krämer et al. (2016, their Fig. 4)).* '

Please  briefly explain  the aerosol classifiaction.

In addition, why not define two classes of  supersaturation,
  $HET^{(MD)}$_in/out    and     $HET^{(CS)}$_in/out  ?  This would provide even more detail!

7)  Page 6 line15:

'…; *low clouds are depicted in yellow.*'   green *??*

**Technical recommendations:**

- Fig. 1:  BSR < 2  would be better BSR <> 2  in the scheme

        general:   DEP      why not   ISSR_in      sounds more clear
        general:   ISSR                ISSR_out     sounds more clear

- Fig. 2 b:   explain also the green color

- Fig. 3 - 7:  insert an arrow to show the wind direction

- Fig. 4:     insert a panel with the ECMWF temperature !

---

## Referee Comment (RC2) · Anonymous Referee #3 · 28 Nov 2016

Review of 'Determining stages of cirrus life-cycle evolution: A cloud classification scheme' by B. Urbanek et al. submitted to Atmos. Meas. Tech.

This manuscript describes the use of airborne lidar measurements of water vapor to demonstrate a cirrus classification scheme that is able to identify the evolution stages of the cirrus life-cycle using primarily relativity humidity with respect ice thresholds deemed from the literature. This is a interesting idea that would be useful in the analysis of airborne and remotely sensed cirrus clouds and would put measurements of cirrus microphysical properties in context with cirrus lifecycle. Overall the manuscript is well written and well organized, but I have some concerns about the data as presented. First, the uncertainty in the computed relative humidity measurement is only loosely mentioned and should be considered in the context of the classification scheme. Second, I have some concerns about the generating mechanism described in this case.

I maybe misinterpreting the data as it is represented in the figures. This may influence the lifecycle, formulation, and interpretation of the classification scheme. These can likely be easily addressed by clarifications from the authors. I recommend this manuscript to be published after these concerns are addressed.

Specific Comments

p. 2 L19-25: Heymsfield 1975 was the first to illustrate and document the vertical and dynamical structure of cirrus uncinus clouds and should be stated as such in this paragraph. I am not sure I would agree with the statement '. . .where individual clouds may show strongly different distributions.' It is the cloud type or generating mechanism that will influence this distribution (i.e. anvil, synoptic, in situ generated, orographic). There have been a number of studies (Sassen, Mace, Protat) that have looked at these differences.

p. 2 L19 and p. 7, L27: The Comstock et al. 2004 study was for data over Oklahoma, U.S.A., not France.

p. 4 L20: The stated uncertainty in the relative humidity with respect to ice (RHI) is 10-15%, which is quite large. Your proposed classification scheme relies on thresholds for RHI (Fig. 1, Table 1). There is currently no discussion of how this uncertainty impacts the classification scheme. Clearly a 10-15% uncertainty would have quite an impact on this thresholding approach. This is a major issue that needs addressing.

Vertical velocity plays a key role in the initiation and evolution of cirrus clouds. Did you consider including the in situ vertical velocity measurements from the HALO in your classification scheme? This could help compensate for the errors in the RHI data.

p. 5 L32: It is stated that high dust concentrations were expected. What is the evidence for this statement? Did you look at the lidar ratios or the in situ aerosol data? Please provide some quantitative justification for this statement. Later, depolarization ratio is used (P. 6, L25) but still there is no reference, nor are typical values of dust mentioned.

[Figure]

Fig. 2b - SEVIRI image: Can you provide the time at the northern most point on the white flight track line and also the southern most point? This will help put the lidar profiles in context with the satellite data (I assume the northern most point is roughly 14:20 UTC). Also, I am wondering what type of cloud the green color is associated with? Are these convective clouds? The reason I ask is that the lidar profiles in Fig. 3, 4, and 5 suggest that the clouds are thicker (3-4 km deep) than what I would expect in orographic lee clouds (∼1-2 km deep) and the cloud looks to be more stratiform in its morphology. This suggests to me the cloud is a cirrostratus or anvil, that may have some gravity wave influences from the mountains. This would influence your discussion and conclusions about the formation and evolution of the cloud since anvil cirrus is formed in a much different environment than synoptic or in situ/orographic cirrus. Please consider the possibility if I am interpreting the figure appropriately. Adding a brightness temperature scale to the SEVIRI image would be helpful.

P. 7 L25: '...probably caused by large-scale descent.' The ECMWF vertical velocity data could provide some clues to the large-scale dynamics.

Editorial: p. 2 L16: loose -> lose p. 2 L30: ascends -> ascent p. 5 L14: 'a uncertainty' -> 'an uncertainty' p. 5 L18: represent -> represents
* * *

---

## Author Comment (AC1) · 22 Dec 2016

For the full response to RC1 and RC2 and a version of the manuscript highlighting the changes, see the supplement .zip archive.

Please also note the supplement to this comment:
http://www.atmos-meas-tech-discuss.net/amt-2016-332/amt-2016-332-AC1-supplement.zip

[Figure]

**Author's final response**

"Determining stages of cirrus evolution: A cloud classification scheme" by B. Urbanek et al.

**Review RC1 by Anonymous Referee #1**

We thank Referee #1 for carefully reading our manuscript and for the suggestions that helped us to improve our work. In the following we will answer to his specific comments.

**Introduction from Referee**

The authors present an attempt to determine the stages of cirrus life-cycle evolution based on in-cloud RHi measurements performed by the airborne Lidar WALES. Though I like the idea and also find the paper well organized and fluently written, I have a major concern with respect to the proposed cirrus life-cycle classification scheme which I explain in the following. To my opinion this point should be cleared before publishing the manuscript in ACP.

**Comment 1 from Referee (Major comment)**

In the introduction, the authors state:

'*In order to gain more insight into the particular role of different cirrus clouds, great efforts were made to classify cirrus by the meteorological contexts in which they occur (Jackson et al., 2015; Muhlbauer et al., 2014). Categories include "synoptic", "orographic", "lee wave" and "anvil" cirrus. Recently Krämer et al. (2016) introduced a more general classification distinguishing the groups of "liquid origin" and "in situ" clouds that describe whether the cirrus formed from a pre-existing liquid cloud or from cloud-free air. Such a classification of recorded data is a prerequisite for statistically investigating the specific properties and influences of different clouds, and to extract the governing mechanisms and parameters from remote sensing and in situ measurements.*'

However, the cirrus life-cycle classification scheme presented in the paper holds only for 'in situ' formed cirrus clouds. In the so-called 'liquid origin' cirrus, the meaning of 'SUB' will be similar, but what about the interpretation of 'DEP', HETin and HOMin in case of pre-existing ice? It is very likely that in case of further lifting of a liquid origin cirrus cloud the supersaturation rises to values of DEP, HETin or HOMin (then, a new, homogeneous nucleation event can occur on top of the liquid origin cirrus), but they are at different stages of cirrus evolution than the in situ cirrus.

In a recent publication of Wernli et al. (2016), GRL, the frequencies of occurrence of in situ and liquid origin cirrus are analyzed from 12 years of ERA-Interim ice clouds in the North Atlantic region. Wernli et al. found that: 'Between 400 and 500 hPa more than 50% are liquid-origin cirrus, whereas this frequency decreases strongly with altitude (<10% at 200hPa).'

Thus, it seems to be important that first of all these two types of cirrus can be identified by a cirrus classification scheme before going in the detail of stages of cirrus life-cycle evolution. So I would highly encourage the authors to continue their work by including an analysis of the cirrus origin prior to the investigation of the stages of evolution. It might be

**Fig. 1.** Preview of the authors' final response found in supplement .zip archive

**Determining stages of cirrus  evolution: A cloud classification scheme**

Benedikt Urbanek[1], Silke Groß[1], Andreas Schäfler[1], and Martin Wirth[1]

[1]Deutsches Zentrum für Luft- und Raumfahrt, Institut für Physik der Atmosphäre, Oberpfaffenhofen, Germany

*Correspondence to:* Benedikt Urbanek (benedikt.urbanek@dlr.de)

**Abstract.** Cirrus clouds impose high uncertainties on climate prediction, as knowledge on important processes is still incomplete. For instance it remains unclear how cloud microphysical and radiative properties change as the cirrus evolves. Recent studies classify cirrus clouds into categories including "in situ", "orographic", "convective" and "liquid origin" clouds and investigate their specific impact. Following this line, we present a novel scheme for the classification of cirrus clouds that addresses the need to determine specific stages of cirrus  evolution. Our classification scheme is based on airborne Differential Absorption and High Spectral Resolution Lidar measurements of atmospheric water vapor, aerosol depolarization, and backscatter, together with model temperature fields and simplified parameterizations of freezing onset conditions. It identifies regions of supersaturation with respect to ice (ISSR), heterogeneous and homogeneous nucleation, depositional growth, and ice sublimation and sedimentation with high spatial resolution. Thus all relevant stages of cirrus evolution can be classified and characterized. In a case study of a gravity lee wave influenced cirrus cloud, encountered during the ML-CIRRUS flight campaign, the applicability of our classification is demonstrated. Revealing the structure of cirrus clouds, this valuable tool might help to examine the influence of  evolution stages on the cloud's net radiative effect and to investigate the specific variability of optical and microphysical cloud properties in upcoming research.

**1 Introduction**

Cirrus play an important role for weather and climate: besides their influence on the water vapor budget in the upper troposphere through condensation and evaporation (Dinh et al., 2014) and dynamics due to latent heat (Spichtinger, 2014), they modify the radiation balance of the Earth and atmosphere. Thin, opaque cirrus clouds transmit most of the incident solar radiation and absorb long-wave radiation from the Earth's surface. As they are typically high and cold, they only emit little long-wave radiation into space, and thus cause a trapping of radiative energy in the Earth-atmosphere system, which eventually contribute to a rising surface temperature. If the cloud is thick, reflection of solar radiation back to space can get greater than the long-wave absorption, and consequently can cause the surface of the Earth to cool (Baran, 2009). This net radiative effect depends on macroscopic cloud properties like optical thickness, ice water content, and geometric extent as well as on its microphysical parameters such as ice crystal number, size, and shape (Schnaiter et al., 2016; Gallagher et al., 2012; Zhang et al., 1999).

**Fig. 2.** Preview of the changed manuscript found in supplement .zip archive

---

## Referee Comment (RC3) · Anonymous Referee #2 · 8 Jan 2017

Review of
**Determining stages of cirrus life-cycle evolution: A cloud classification scheme**
by Urbanek et al.

**General comment:**
In this study a classification scheme for stages of cirrus cloud life cycle is presented. The scheme is based on LIDAR data in combination with meteorological data (temperature and pressure) from ECMWF. In a case study of orographic cirrus clouds as measured during the ML-CIRRUS campaign the scheme is applied and the results are interpreted.

Generally this is an interesting and important contribution to ice cloud research; thus, this study is an appropriate contribution for AMT. However, there are some issues, which must be clarified before this manuscript can be accepted for publication. Therefore, I recommend major revisions of the manuscript. In the following I will explain my concerns in details

**Major points**

1. Classification scheme and interpretation of results

   The general aim of the scheme is not really clear to me. I recommend that the authors give a bit more information about the aim and the possible use of the scheme.

   In general, I agree with the discrimination between regions of potential ice nucleation, moderate supersaturation and subsaturation, since this reflects the different thermodynamic states of the system. However, the role of the class HET is not clear to me and seems to cause severe problems:

   (a) Since heterogeneous ice nucleation is not well understood, and ice nucleation on solid particles depends on many details, a general nucleation threshold (as e.g. for homogeneous nucleation, but see minor comment below) cannot be determined. This problem is already reflected in this scheme by the use of 2 different parameterisations and their difference of about 20-30%. Therefore, the definition of the class $HET_{in/out}$ is quite arbitrary, since the lower bound is very fuzzy.

   (b) For cloud free air the class might be useful, since then the possibility of heterogeneous nucleation could be estimated. But again the arbitrary thresholds of heterogeneous nucleation make it very difficult to use this information in a meaningful way.

   (c) For cloudy data, this class might lead to severe misinterpretation of the data. In the text it is suggested that for data points of $HET_{in}$ heterogeneous nucleation takes place or even ice crystals in this category stem from heterogeneous nucleation. This suggestion is not correct because of the problem stated in (a): The nucleation threshold is not well-posed, thus it might be that using a low threshold no heterogeneous nucleation takes place (since the IN need higher supersaturation); thus, the interpretation of ongoing nucleation would be wrong. In the case study the lower threshold is used, but it is not clear if this is really the right one.

   These problems weaken the classification scheme in a serious way; therefore I recommend either to remove the class $HET_{in}$ completely or even to refine the representation using different heterogeneous nucleation thresholds as a standard. Perhaps additional information could be given in addition to the coarse classification HET. If the class HET is kept in the scheme, its use, benefits and problems should be described carefully.

There is another issue regarding the interpretation of the classification. The scheme is based on measurements, i.e. on an Eulerian viewpoint, since the time evolution cannot be seen. If ice crystals were found in the class $\mathrm{HET_{in}}$, they are not necessarily formed by heterogeneous nucleation. The classification just can tell some information of the actual state of possible nucleation, but not about the particles, which are already in the air mass. For instance, sedimenting ice crystals could be found in the air mass, but they were formed at higher altitudes under completely different conditions. The authors should mention this problem, since confusing Lagrangian and Eulerian viewpoint could lead to completely wrong results.

2. Analysis of case study

The demonstration of the classification scheme was carried out using a very special case of orographic cirrus clouds. In general this is ok, but the interpretation of the case could be more specific.

(a) Probably weak sedimentation
Since the cirrus cloud was obviously formed by a (strong) wave, probably sedimentation was not a big issue, since many small ice crystals were formed. The region at the top of the cloud showing very high backscatter ratios is a hint into this direction. Maybe the authors could use the analysis of the trajectory in order to estimate the vertical velocities, which might be interesting for homogeneous nucleation.

(b) Descent of the cloud
The authors claim that the descent of the ice cloud is probably triggered by large-scale downdrafts. However, this could be corroborated using ECMWF wind data, which are available; this would also strengthen the argument for the occurrence of region DEP and SUB. In addition, they should estimate sedimentation velocities of ice crystals for typical sizes in order to rule out the case of sedimenting ice crystals leading to this cloud descent.

(c) High supersaturation without ice nucleation
In the measurement time 14:34-14:36 high ice supersaturation occurs (at least higher than $RHi_{\mathrm{het}}$) without ice nucleation. This might point to the possibility that either heterogeneous nucleation at high thresholds or even only homogeneous nucleation are the preferred nucleation types in this situation. Again, this points to the weakness of the definition of HET regions without a concise threshold for heterogeneous nucleation. What about measurement errors in relative humidity (of order of 10-15%)? Might it be possible to reach higher values of RHi?

**Minor points:**

1. Page 2, lines 8-12: in situ vs. liquid origin ice crystals
The discrimination between these two types is based on thermodynamics

- liquid origin: freezing of existing water droplets at water saturation
- in situ: freezing of solution droplets or heterogeneous nucleation at ice supersaturation but below water saturation

Maybe this could be mentioned in the text. Please also add the reference Wernli et al. (2016), since the classification (in situ/liquid origin) is also used in this study.

2. Page 2, lines 19-25: vertical structure of ice clouds
The description is probably only valid for stratiform cirrus clouds, formed by in situ formation

mechanisms. For liquid origin ice clouds and for clouds with strong dynamics (waves or instabilities) the structure might be different. This should be mentioned in the text.

3. Use of ECMWF data
   Which kind of ECMWF data is used and how? Is there a mixture of analysis data (available every 6 hours) with short term forecasts? Please explain this in more details.

4. Measurements of temperature during ML-CIRRUS
   As far as I remember, during ML-CIRRUS temperature profiles were measured with the MTP instrument. Why do you not use these measurements instead of coarse resolution ECMWF data?

5. Page 4, lines 32-33: Reference for sufficient amount of solution droplets
   A suitable reference for the occurrence of sufficient solution droplets, i.e. sufficient soluble aerosol particles, would be Minikin et al. (2003).

6. Page 5, lines 1-5: Representation of homogeneous nucleation
   The representation of homogeneous freezing of solution droplets and the derivation of freezing thresholds is very short and misleading for non-experts; it should be expanded. The volume nucleation rate depends on water activity, i.e. $J = J(\Delta a_w) = J(RHi, T)$ and the nucleation rate $\omega$ is composed by using the volume of a solution droplet $V = \frac{4}{3}\pi r_0^3$ with a size $D = 2r_0$, i.e. $\omega = JV$. Koop et al. (2000) made the (arbitrary) setting of $\omega = 1\,\mathrm{min}^{-1}$, which means that all solution droplets of radius $r = r_0$ freeze within a timestep of one minute ($\Delta t = 1\,\mathrm{min}$) with a probability of $P = 1 - \exp(-\omega\Delta t) \approx 0.63$. However, the choice of $\omega$ is quite arbitrary and should be mentioned, while $D = 2r_0 = 0.5\,\mu\mathrm{m}$ might be a reasonable choice of a typical size. This should be mentioned in the text.

7. Page 9, line 4: Gravity waves are not really small-scale dynamics
   The statement of gravity waves as small-scale dynamics is a bit weird and should be rewritten; maybe mesoscale dynamics is a better classification, since small scale is more associated with turbulence.

**Technical comments**

1. The colour bars in almost all figures are not easy to read. Especially for figures 4 and 6, colour bars with more colours and/or clearer increments should be used.

2. In figure 5, the difference between regions HET and HOM cannot be seen, since the colours are too similar.

3. In figure 6 the trajectory could also be shown for clarification of the derivation.

**References**

Minikin, A., Petzold, A., Ström, J., Krejci, R., Seifert, M. and co-authors. 2003. Aircraft observations of the upper tropospheric fine particle aerosol in the Northern and Southern Hemispheres at midlatitudes. Geophys. Res. Lett. 30, 1503. DOI: 10.1029/2002GL016458.

Wernli, H., M. Boettcher, H. Joos, A. K. Miltenberger, and P. Spichtinger, 2016: A trajectory-based classification of ERA-Interim ice clouds in the region of the North Atlantic storm track. Geophys. Res. Lett., 43, 6657-6664, doi:10.1002/2016GL068922.

---

## Author Comment (AC2) · 22 Feb 2017

For the full response to RC3 and a version of the manuscript highlighting the changes, please see the supplement .zip archive.

Please also note the supplement to this comment:
http://www.atmos-meas-tech-discuss.net/amt-2016-332/amt-2016-332-AC2-supplement.zip
* * *
[Figure]

**Author's final response**

**Review RC3 by Anonymous Referee #2**

We thank Referee #2 for his suggestions to improve the quality of our work. In the following we will answer to his specific comments.

**General Comment from Referee**

In this study a classification scheme for stages of cirrus life-cycle is presented. The scheme is based on LIDAR data in combination with meteorological data (temperature and pressure) from ECMWF. In a case study of orographic cirrus clouds as measured during the ML-CIRRUS campaign the scheme is applied and the results are interpreted.

Generally this is an interesting and important contribution to ice cloud research; thus, this study is an appropriate contribution for AMT. However, there are some issues, which must be clarified before this manuscript can be accepted for publication. Therefore, I recommend major revisions of the manuscript. In the following I will explain my concerns in details.

**Comment 1 from Referee (Major comment)**

Classification scheme and interpretation of results

The general aim of the scheme is not really clear to me. I recommend that the authors give a bit more information about the aim and the possible use of the scheme.

In general, I agree with the discrimination between regions of potential ice nucleation, moderate supersaturation and subsaturation, since this reflects the different thermodynamic states of the system. However, the role of the class HET is not clear to me and seems to cause severe problems:

(a) Since heterogeneous ice nucleation is not well understood, and ice nucleation on solid particles depend on many details, a general nucleation threshold (as e. g. for homogeneous nucleation, but see minor comment below) cannot be determined. This problem is already reflected in this scheme by the use of 2 different parameterisations and their difference of about 20-30%. Therefore, the definition of the class $HET_{in/out}$ is quite arbitrary, since the lower bound is very fuzzy.

(b) For cloud free air the class might be useful, since then the possibility of heterogeneous nucleation cloud be estimated. But again the arbitrary thresholds of heterogeneous nucleation make it very difficult to use this information in a meaningful way.

(c) For cloudy data, this class might lead to severe misinterpretation of the data. In the text it is suggested that for data points of $HET_{in}$ heterogeneous nucleation takes place or even ice crystals in this category stem from heterogeneous nucleation. This suggestion is not correct because of the problem stated in (a): The nucleation threshold is not well-posed, thus it might be that using a low threshold no heterogeneous nucleation takes place (since the IN need higher saturation); thus the interpretation of ongoing nucleation would be wrong. In the case study the lower threshold is used, but it is not clear if this is really the right one.

**Fig. 1.** Preview of the authors' response found in supplement .zip archive

**Determining stages of cirrus  evolution: A cloud classification scheme**

Benedikt Urbanek[1], Silke Groß[1], Andreas Schäfler[1], and Martin Wirth[1]

[1]Deutsches Zentrum für Luft- und Raumfahrt, Institut für Physik der Atmosphäre, Oberpfaffenhofen, Germany

*Correspondence to:* Benedikt Urbanek (benedikt.urbanek@dlr.de)

**Abstract.** Cirrus clouds impose high uncertainties on climate prediction, as knowledge on important processes is still incomplete. For instance it remains unclear how cloud microphysical and radiative properties change as the cirrus evolves. Recent studies classify cirrus clouds into categories including "in situ", "orographic", "convective" and "liquid origin" clouds and investigate their specific impact. Following this line, we present a novel scheme for the classification of cirrus clouds that

5 addresses the need to determine specific stages of cirrus  evolution. Our classification scheme is based on airborne Differential Absorption and High Spectral Resolution Lidar measurements of atmospheric water vapor, aerosol depolarization, and backscatter, together with model temperature fields and simplified parameterizations of freezing onset conditions. It identifies regions of supersaturation with respect to ice (ISSR), heterogeneous and homogeneous nucleation, depositional growth, and ice sublimation and sedimentation with high spatial resolution. Thus all

10 relevant stages of cirrus evolution can be classified and characterized. In a case study of a gravity lee wave influenced cirrus cloud, encountered during the ML-CIRRUS flight campaign, the applicability of our classification is demonstrated. Revealing the structure of cirrus clouds, this valuable tool might help to examine the influence of  evolution stages on the cloud's net radiative effect and to investigate the specific variability of optical and microphysical cloud properties in upcoming research.

15 **1  Introduction**

Cirrus play an important role for weather and climate: besides their influence on the water vapor budget in the upper troposphere through condensation and evaporation (Dinh et al., 2014) and dynamics due to latent heat (Spichtinger, 2014), they modify the radiation balance of the Earth and atmosphere. Thin, opaque cirrus clouds transmit most of the incident solar radiation and absorb long-wave radiation from the Earth's surface. As they are typically high and cold, they only emit little long-wave

20 radiation into space, and thus cause a trapping of radiative energy in the Earth-atmosphere system, which eventually contribute to a rising surface temperature. If the cloud is thick, reflection of solar radiation back to space can get greater than the long-wave absorption, and consequently can cause the surface of the Earth to cool (Baran, 2009). This net radiative effect depends on macroscopic cloud properties like optical thickness, ice water content, and geometric extent as well as on its microphysical parameters such as ice crystal number, size, and shape (Schnaiter et al., 2016; Gallagher et al., 2012; Zhang et al., 1999).

**Fig. 2.** Preview of changed manuscript found in supplement .zip archive